# The association between the use of video games, social media and online dating sites, and the symptoms of anxiety and/or depression in adults aged 25 and over

anxiety; depression; internet use; adults; cross-sectional study; French cohort

**Corresponding author:**
Murielle Mary-Krause;
Email: murielle.mary-krause@iplesp.upmc.fr

Murielle Mary-Krause is co-last author.

Maria El Haddad, Irwin Hecker, Solène Wallez, Murielle Mary-Krause  and Maria Melchior 

Sorbonne Université, INSERM, Institut Pierre Louis d'Epidémiologie et de Santé Publique (IPLESP), Equipe de Recherche en Epidémiologie Sociale (ERES), F75012 Paris, France

## Abstract

People tend to spend more time in front of their screens, which can have repercussions on their social life, physical and mental health. This topic has mainly been studied in adolescents. Therefore, our study tested associations between the use of video games, social media and online dating leading to sexual relations (ODLSR), and symptoms of anxiety and/or depression among adults aged 25 and over. Data from the 2018 TEMPO cohort study were analyzed (n = 853, 65.0% women, aged 25–44, with an average of 37.4 ± 3.7 years). The exposure variables were as follows: (a) the frequency of video game use, (b) time spent on social media and (c) ODLSR. Data were analyzed using multivariate logistic regression models, adjusted for participants' sociodemographic characteristics as well as history of mental health problems. Among the participants, 8.6% presented symptoms of anxiety and/or depression. An association between ODLSR and symptoms of anxiety and/or depression was found, especially among women. The results of this study will facilitate the improvement of support and care for adults, especially those with symptoms of anxiety and/or depression using dating applications. Future studies should investigate the determinants of using online meeting websites and their relationship with the occurrence of psychological difficulties in longitudinal studies to establish causality.

## Impact statement

In recent years, and even more during the COVID-19 pandemic, screen use in various forms has become a frequent leisure activity among children, adolescents and adults in industrialized countries. The use of screen-based media has significant consequences on people's lives, affecting sociability, physical activity, dietary intake and psychoactive substance use. Recently, concerns have grown about the impact of screen-based media use on individuals' mental health. However, the majority of studies on video games and social media have focused on adolescents or young adults, with some on the elderly, but very few have evaluated associations with mental health among adults aged 25 and over. Thus, the objective of our study was to assess the relationship between video games, social media and online dating website use with symptoms of anxiety and depression in adults aged 25 and over, controlling for sociodemographic characteristics, as well as preexisting mental health problems. The findings from this study will contribute to a deeper understanding of this less-studied adult population, and will help improve the management of individuals who suffer from mental health disorders and engage with these applications. This is particularly important in a society where adults over 25 also tend to use screens widely – both professionally and recreationally – and are at risk of experiencing symptoms of anxiety and/or depression.

Cambridge Prisms

CAMBRIDGE UNIVERSITY PRESS

## Introduction

In recent years, in industrialized countries, screen usage in various forms has become a common leisure activity among children, adolescents and adults (Global Web Index, 2018). In a large number of European countries, the percentage of the population using the Internet exceeds 90%, with the European average being 89.4%, and France ranking as the second European country for the Internet usage after Germany (Internet World Stats, 2022). In France, 98% of French adults aged 25–39 are connected to the Internet, and the time spent on social media has increased globally (Baromètre du numérique, 2019; Global Web Index, 2018). In 2021, an average of 1 h and

46 min was spent daily on social media in France (Gaudiaut, 2022). According to a survey conducted in 2015 by the French National Center for Cinema and Animated Image, 7 out of 10 French people play video games (73.3%), with 80.3% of players being adult (41.6% aged 15–34 and 23.9% aged 35–49). Half of the video game users play on a daily basis, and the majority are men (Centre National du Cinéma et de l'image animée, 2015). Moreover, dating sites and apps are becoming more numerous. Few data about dating website use among French adults in the general population exist in research articles. One study conducted in 2013–2014 showed that 14% of 26–65-year-olds had used dating sites in 2013, compared to 9% in 2006, with the percentage decreasing with age from 29% among 26–30-year-olds to 3% among 61–65-year-olds (Bergström, 2016). A more recently published study conducted in 2018 in Normandy, France, among 1,208 teenagers aged 15–17 years showed that the weighted prevalence of active cybersexuality, including the use of dating websites, was 60% (Rousseau et al., 2023). However, this study only included teenagers, and their prevalence of active cybersexuality must differ significantly from that of much older adults. Moreover, active cybersexuality in this study encompassed more than just dating website use. When studies are carried out in adults, they often concern special populations, such as people living with HIV (Jacomet et al., 2020).

The use of screen-based media has consequences on people's lives. One of the direct consequences can be on sociability, as spending time on the Internet limits face-to-face social interactions (Lissak, 2018; Small et al., 2020). Nevertheless, it has been suggested that connecting with others online is a new form of sociability, which allows to have larger networks and easy communication (Fortunati et al., 2013). In particular, online dating websites and applications can sometimes replace spontaneous encounters, as they are easy to use and meeting people directly in "real life" can seem simply frightening. Likewise, time spent on video games and social media could replace time and activities spent with friends or family, including physical activities and result in loneliness (van den Eijnden et al., 2018; Alshehri and Mohamed, 2019). Moreover, screen time can be related to harmful health behaviors such as sedentary lifestyle, obesity, sleep disorders, eye disorders and addictive behaviors (Melchior et al., 2014; Biddle et al., 2017; Cabré-Riera et al., 2019; Jaiswal et al., 2019).

Recently, concerns have grown about the impact of screen-based media use on individuals' mental health (Kaess et al., 2014; Brailovskaia and Margraf, 2018; Ioannidis et al., 2018; Stockdale and Coyne, 2018; Brailovskaia et al., 2019b; Holtzhausen et al., 2020; Cannito et al., 2022). In particular, there is evidence that internet use or high screen time can be related to adolescents' symptoms of depression, anxiety, suicidal ideation and suicide attempts (Liu et al., 2016). In addition, young adults with high levels of rejection or with some mental health problems such as low self-esteem, or even severe mental health conditions were more likely to engage in online dating (Blackhart et al., 2014; Hance et al., 2017; Strubel and Petrie, 2017; Rydahl et al., 2021). Moreover, a meta-analysis of Marino et al. among 13,929 participants with an average age of 21.93 years (range: 16.5–32.4) confirmed a positive correlation between problematic Facebook use and psychological distress, as well as a negative correlation between problematic Facebook use and well-being (Marino et al., 2018). However, the classification and criteria to define internet addiction are still controversial and subject to debate (Poli, 2017). Few studies have been carried out on the consequences of internet use among adults, with the majority focusing on the elderly population. Some studies have found a reduction in depressive symptoms and suicidal ideation (Jun and

Kim, 2017; Wang et al., 2019) as well as an increase in life satisfaction (Lifshitz et al., 2018). Unlike young people, internet use enables seniors to enjoy greater social inclusion (Forsman and Nordmyr, 2017). Moreover, the majority of studies on video games and social media focus on adolescents or young adults, and only a few studies have evaluated the associations with mental health among adults aged 25 and over. Some studies were carried out among specific vulnerable populations, such as inpatients of a psychosomatic rehabilitation clinic (Brailovskaia et al., 2019a). Others were conducted during the COVID-19 pandemic (Gao et al., 2020; Brailovskaia and Margraf, 2022) a period when social media use increased (Masaeli and Farhadi, 2021; Zhao and Zhou, 2021).

Thus, the objective of our study was to assess the relationship between video game usage, social media engagement and online dating website use with symptoms of anxiety and/or depression in adults, controlling for sociodemographic characteristics as well as preexisting mental health problems that could potentially confound this association. Our hypothesis was that there exists an association, even among adults, between video game usage, social media engagement and online dating website use, and symptoms of anxiety and/or depression.

## Methods

### Sample and procedure

The TEMPO study (Epidemiological Trajectories in Population – "Trajectoires ÉpidéMiologiques en Population") is a French longitudinal observational cohort that aims to understand lifecourse trajectories of mental health and addictive behaviors (including tobacco, alcohol, cannabis or other illicit drugs) from adolescence to adulthood (Mary-Krause et al., 2021).

This cohort started in 1991 among 2,582 persons aged 4–18 years, randomly drawn from the offspring of GAZEL cohort participants, an epidemiological cohort study comprising 20,000 volunteers employed by France's national utilities company (Goldberg et al., 2015). TEMPO participants were followed-up in 1999, 2009, 2011, 2015 and 2018 (Supplementary Figure 1). To counterbalance attrition, the sample was supplemented in 2011 by recruiting participants aged 18–35, whose parents also participated in the GAZEL cohort. Overall, 3,401 individuals participated in at least one TEMPO cohort study assessment between 1991 and 2018.

The present study is based on data from participants who, after being informed about the study's purposes and agreeing to participate, completed the 2018 TEMPO cohort assessment (n = 864, 71% participation rate), which included questions about screen and media use.

The TEMPO cohort received approval from the ethical data collection supervisory bodies in France, including the Advisory Committee on the Treatment of Information for Health Research (Comité consultatif sur le traitement de l'information en matière de recherche dans le domaine de la santé, CCTIRS), and the French national committee for data protection (Commission Nationale de l'Informatique et des Libertés, CNIL, no. 908163).

### Measures

#### Study outcome: Symptoms of anxiety and/or depression

Symptoms of anxiety and/or depression were measured using the adult self-report (ASR) (Achenbach et al., 2003), which includes 41 items on symptoms of anxiety and/or depression over the preceding 12 months. The ASR is a validated standardized self-

administered questionnaire assessing different dimensions of mental health across different age groups, designed to measure symptoms, that may be indicative of psychiatric disorders (Rescorla and Achenbach, 2004). Studying symptoms has been found to be a valid indicator of disorders, and also provides optimal statistical power for the statistical analyses (Waszczuk et al., 2017). The anxious and depressed scale of the ASR has high reliability with test–retest correlations of 0.87, high internal consistency with a Cronbach's alpha of 0.88, and high validity with a cross-validated percent of adults correctly classified as referred vs. non referred equal to 87% (sensitivity = 80%, specificity = 95%) (Achenbach and Rescorla, 2003). The score of symptoms of anxiety and/or depression was calculated by summing all items and dichotomizing at or above the 85th percentile, which identifies individuals with potentially clinically significant symptoms (Achenbach et al., 2003).

### Exposure variables

Three exposure variables were studied: the frequency of playing video games, time spent on social media and online dating.

The frequency of playing video games was assessed using the following question: "In the past 6 months, how often have you played video games (on a computer, console, cell phone or tablet)?" The response options were: Never, once per month, between one and four times per month, several times a week and every day. No standard cut-off point was available for the frequency of playing video games. Therefore, the answers were grouped according to the variable distribution and divided into three categories: "Never", "one to four times per month" and "multiple times per week".

Social media use was assessed using the following questions: "If you have an account on the following social media or exchange platforms (Facebook, LinkedIn, YouTube, Pinterest, Google+, Copains d'Avant (a website making it possible to identify former classmates), Instagram, Viadeo, Snapchat, Twitter, Flickr, Tumblr, Myspace or Other), how many hours per day, per week, per month or per year do you spend using it?". There was no standard cut-off point available for the time spent on social media among adults. However, it is recommended that the time spent seated should not exceed two consecutive hours to avoid sedentary lifestyle (Ministère de la santé et de la prévention, 2022). Furthermore, a study conducted among adolescents by AlSayyari et al. used a cut-off point of 2 h per day (AlSayyari and AlBuhairan, 2018). In the current study, the time spent on social media was estimated in hours per day, with an average of 1.5 h (standard deviation [SD] = 2.8). Therefore, a cut-off point of 2 h per day was chosen based on the distribution of the variable, resulting in the following categories: "Not at all", "2 h per day or less" and "more than 2 h per day".

Regarding online dating websites, participants answered the following question: "Have you ever had sexual relations with someone you met online on a dating website?" with answer choices: "Yes, once", "Yes, several times", "No, never", "I do not wish to answer". As only four people chose not to answer the question, making it impossible to create a separate category in the regression models, these four individuals were considered as missing variable in the regression models, and the variable was dichotomized into "Yes" and "No". In this article, we will refer to this variable as "online dating leading to sexual relations" (ODLSR).

### Covariates

Covariates included participants' age, sex, living situation, same-sex sexual relations, socioeconomic index (SEI) and prior history of symptoms of anxiety and/or depression. As assessed in previous studies conducted among the TEMPO cohort participants, age was dichotomized into the two following categories: "under 30″ and "30 and above". This allows us to account for the difference between young adults and older adults (Lachman, 2004; Barry et al., 2022). Living situation was defined as "lives with a partner and children", "lives with a partner without children" or "lives alone".

History of same-sex sexual relations was assessed by the following question: "Have you ever had sexual relations with someone of the same sex as you?" to which participants could answer "Yes, in the past 12 months", "Yes, but not in the past 12 months", "No, never" or "I do not wish to answer". Answers were dichotomized into "yes", "no" and "do not wish to answer".

Participants' SEI level was determined by combining four variables: educational level (< Bachelor's degree +3 versus ≥ Bachelor's degree +3), occupational grade (manual workers/clerks vs. other occupations), job stability over the previous 12 months (unstable vs. stable employment) and lifetime unemployment (≤ 6 months vs. >6 months). Each of these variables was coded as 0 and 2, respectively, and they were then summed to obtain an overall index. The lowest quartile was categorized as "low SEI" versus "intermediate-high SEI" (Redonnet et al., 2012).

Information on the history of symptoms of anxiety and/or depression was assessed using the ASR (Achenbach et al., 2003) and obtained from TEMPO cohort waves preceding 2018, that is, 2011 or 2009, with the most recent information being taken into account. The score of symptoms of anxiety and/or depression was calculated by summing all items and dichotomizing at or above the 85th percentile, which identifies individuals with potentially clinically significant symptoms (Achenbach et al., 2003).

Social support was assessed by determining the number of family members and friends that participants felt close to, using the following questions: "How many of your family members do you feel close to (i.e., you can talk to them about personal things or you can call for help)?" – "How many close friends do you have?" with answer choices "none", "1 or 2", "3 to 5", "6 to 9" and "10 or more". The last two categories were grouped together for the analysis.

The analysis also considered whether participants felt they required more assistance from those around them and from their partner, using the following questions: "Over the past 12 months, would you have needed more help from those around you?" – "In the last 12 months, would you have needed more help from your partner?" Participants could respond with "Not at all satisfied", "Not completely satisfied", "Neither satisfied nor dissatisfied", "Satisfied", "Very satisfied" or "Do not have a partner" for the second question. A summary variable regarding the need for more help was created.

### Statistical analysis

To test the associations between different types of screen use and symptoms of anxiety and/or depression, we first conducted bivariate logistic regression. All covariables significantly associated with high levels of symptoms of anxiety and/or depression with a $p$-value ≤0.20 in this bivariate logistic regression model were included in various multivariable logistic regression models. These models included characteristics of screen use both separately and all together, in relation to participants' symptoms of anxiety and/or depression. In additional analyses, we examined interactions between ODLSR and, (a) participants' sex, and (b) history of same-sex sexual relations, and symptoms of anxiety and/or depression. A $p$-value <0.05 was considered to indicate statistical significance. All statistical analyses were performed using SAS® 9.4 software.

## Results

Table 1 summarizes the participants' characteristics. After excluding participants with missing values for symptoms of anxiety and/or depression, the analytical sample size was n = 853, and among them, 8.6% presented symptoms of anxiety and/or depression. Study participants were 25–44 years old (mean = 37.4, SD = 3.7). The majority were female (65.0%), living with a partner and children (64.6%), had never had same-sex sexual intercourse or refused to answer (91.6%), had an intermediate-high SEI level (71.9%), and 31.2% had a history of symptoms of anxiety and/or depression. Although between 45 and 50% of the participants had three to five close friends or family members, 6.7% felt they needed more help from their partner, relatives and friends.

**Table 1.** Characteristics of TEMPO cohort study participants according to presence or not of symptoms of anxiety and/or depression (N = 853, France, 2018)

| Total n (%) | | Symptoms of anxiety and/or depression | | |
| --- | --- | --- | --- | --- |
| | | No n (%) | Yes n (%) | *p*-value[a] |
| Frequency of video game use | | | | |
| Never | 327 (39.0) | 299 (91.4) | 28 (8.6) | 0.127 |
| One to four times per month | 235 (28.0) | 222 (94.5) | 13 (5.5) | |
| Several times per week | 276 (32.9) | 247 (89.5) | 29 (10.5) | |
| Time spent on social media per day | | | | |
| Not at all | 95 (11.2) | 89 (93.7) | 6 (6.3) | 0.705 |
| 2 h or less | 653 (76.6) | 595 (91.1) | 58 (8.9) | |
| More than 2 h | 104 (12.2) | 95 (91.4) | 9 (8.7) | |
| Online dating leading to sexual relations | | | | |
| No | 664 (78.9) | 617 (92.9) | 47 (7.1) | 0.004 |
| Yes | 174 (20.7) | 148 (85.1) | 26 (14.9) | |
| Do not wish to answer | 4 (0.5) | 4 (100.0) | 0 (0.0) | |
| Age | | | | |
| 30 years old or above | 826 (96.8) | 756 (91.5) | 70 (8.5) | 0.630 |
| under 30 years old | 27 (3.2) | 24 (88.9) | 3 (11.1) | |
| Gender | | | | |
| Female | 554 (65.0) | 494 (89.2) | 60 (10.8) | 0.001 |
| Male | 299 (35.0) | 286 (95.7) | 13 (4.4) | |
| Living circumstances | | | | |
| Alone | 185 (21.9) | 159 (86.0) | 26 (14.0) | 0.003 |
| With a partner and children | 545 (64.6) | 511 (93.8) | 34 (6.2) | |
| With a partner without children | 114 (13.5) | 102 (89.5) | 12 (10.5) | |
| History of same-sex sexual relations | | | | |
| No or does not wish not to answer | 766 (91.0) | 703 (91.8) | 63 (8.2) | 0.537 |
| Yes | 71 (8.4) | 63 (88.7) | 8 (11.3) | |

*(Continued)*

**Table 1.** *(Continued)*

| Total n (%) | | Symptoms of anxiety and/or depression | | |
| --- | --- | --- | --- | --- |
| | | No n (%) | Yes n (%) | *p*-value[a] |
| Do not wish to answer | 5 (0.6) | 5 (100.0) | 0 (0.0) | |
| Socioeconomic index | | | | |
| Low | 238 (28.1) | 204 (85.7) | 34 (14.3) | <0.001 |
| Intermediate – high | 609 (71.9) | 571 (93.8) | 38 (6.2) | |
| Prior history of symptoms of anxiety and/or depression | | | | |
| No | 587 (68.8) | 552 (94.0) | 35 (6.0) | <0.001 |
| Yes | 266 (31.2) | 228 (85.7) | 38 (14.3) | |
| Number of close family members | | | | |
| None | 26 (3.1) | 19 (73.1) | 7 (26.9) | 0.001 |
| One or two | 235 (27.7) | 211 (89.8) | 24 (10.2) | |
| Three to five | 423 (49.9) | 391 (92.4) | 32 (7.6) | |
| Six or more | 163 (19.2) | 155 (95.1) | 8 (4.9) | |
| Number of close friends | | | | |
| None | 32 (3.8) | 25 (78.1) | 7 (21.9) | 0.008 |
| One or two | 174 (20.5) | 158 (90.8) | 16 (9.2) | |
| Three to five | 387 (45.6) | 350 (90.4) | 37 (9.6) | |
| Six or more | 256 (30.2) | 243 (94.9) | 13 (5.1) | |
| Need for more help from your partner, relatives and friends | | | | |
| Yes, a lot more | 57 (6.7) | 40 (70.2) | 17 (29.8) | <0.001 |
| Yes, more | 77 (9.1) | 62 (80.5) | 15 (19.5) | |
| Yes, a little bit more | 220 (25.9) | 195 (88.6) | 25 (11.4) | |
| No, it was sufficient | 496 (58.4) | 480 (96.8) | 16 (3.2) | |

[a]*p*-value of chi-square test or Fisher exact test when one or more expected values are less than 5.

Of the study participants, 39.0% never played video games, while 32.9% were frequent gamers (several times per week), 76.6% spent 2 h a day or less on social media, while 12.2% exceeded 2 h a day on these platforms. Additionally, 20.7% reported having had sexual intercourse with someone they met online. On average, participants spent 1.4 (SD = 2.8) h per day on social media. As shown in Table 2, the time spent on social media was significantly associated with the frequency of video game use (*p* < 0.001) as well as sexual intercourse with a partner met online (*p* < 0.001).

Table 3 presents the unadjusted and adjusted associations between the frequency of video game use, time spent on social media, ODLSR and the presence of symptoms of anxiety and/or depression. In bivariate and multivariate logistic regression analyses (Table 3), compared to never users, video game users and participants who used social media did not show a higher level of symptoms of anxiety and/or depression. However, participants who reported having had sexual intercourse with someone they met online were more than 2 times more likely (95% confidence interval (CI) = 1.15–4.06) to have high levels of symptoms of anxiety and/or depression than those who did not use these websites.

In additional analyses stratified by sex, ODLSR was associated with symptoms of anxiety and/or depression in women (OR = 2.89; 95% CI = 1.41–5.93) but not in men (OR = 0.41; 95% CI = 0.04–4.88).

**Table 2.** Relations between video game use, time spent on social media and sexual intercourse with someone met online a dating website. (N = 853)

| | Frequency of video game use | | | |
| | Never n (%) | One to four times per month n (%) | Several times per week n (%) | p-value[a] |
| --- | --- | --- | --- | --- |
| **Time spent on social media per day** | | | | |
| Not at all | 45 (50.6) | 18 (20.2) | 26 (29.2) | <0.001 |
| 2 h or less | 252 (39.0) | 194 (30.0) | 200 (31.0) | |
| More than 2 h | 30 (29.4) | 22 (21.6) | 50 (49.0) | |
| **Online dating leading to sexual relations** | | | | |
| No | 257 (39.4) | 187 (28.6) | 209 (32.0) | 0.458 |
| Yes | 66 (38.4) | 43 (25.0) | 63 (36.6) | |
| | **Time spent on social media/ day** | | | |
| | Not at all n (%) | 2 h or less n (%) | More than 2 h n (%) | |
| **Online dating leading to sexual relations** | | | | |
| No | 86 (13.0) | 506 (76.2) | 72 (10.8) | <0.001 |
| Yes | 6 (3.5) | 135 (78.0) | 32 (18.5) | |

[a]*p*-value of chi square test.

## Discussion

The objective of our study was to evaluate the association between video game use, time spent on social media and ODLSR, and the likelihood of symptoms of anxiety and/or depression among adults. We found that neither the frequency of video game use nor the time spent on social media were associated with participants' psychological difficulties. However, having met sexual partners online on a dating website was associated with an approximately twofold likelihood of symptoms of anxiety and/or depression, even after accounting for participants' sociodemographic and mental health characteristics.

Unlike our study, which found no significant association between the use of video games or social media and symptoms of anxiety and/or depression, several past studies have shown significant associations between these characteristics among adults and adolescents (Andreassen et al., 2016; Pontes, 2017; Brailovskaia and Margraf, 2018; Yoon et al., 2019; Brailovskaia et al., 2019a). Measures of addictive online behaviors varied according to the studies, with some of them using validated scales that may yield more precise estimates of problematic use than the questions measuring frequency and estimated time of use that we relied upon. However, it is worth noting that most prior studies were conducted among adolescents or young adults, rather than older adults (Yoon et al., 2019). Furthermore, when adults are studied, they are often combined with adolescents (Andreassen et al., 2016; Pontes, 2017; Brailovskaia and Margraf, 2018), and associated factors as well as potential consequences may vary with age. It may be that among adults, as opposed to adolescents, video game and social network use are not associated with mental health. Although his findings showed that higher levels of attention deficit hyperactivity disorder (ADHD) traits were associated with more problematic behavior in video game, Panagiotidi founded no relationship between the frequency and duration of video game play and ADHD traits in an adult population

(Panagiotidi, 2017). Additionally, a meta-analytic study revealed that video game training has positive effects on various cognitive functions, including reaction time, attention, memory and global cognition in very old adults (Toril et al., 2014).

Our findings, indicating an association between the use of online dating websites to have sexual relations and the presence of psychological problems is consistent with previous studies that have reported associations with risk of depression, anxiety and low self-esteem (Strubel and Petrie, 2017; Holtzhausen et al., 2020; Lenton-Brym et al., 2021). Notably, a cross-sectional survey conducted in 2018, which included 437 participants, mainly young adults, found that those who frequently used swipe-based dating applications or used them for an extended period had significantly higher levels of psychological distress and depression (Holtzhausen et al., 2020). People with high levels of rejection sensitivity are especially likely to use online dating websites (Blackhart et al., 2014; Hance et al., 2017). And rejection sensitivity has been associated with mental health disorders, showing stability over time (Gao et al., 2017), and may help explain the association between using online dating websites to have sexual relations and the presence of psychological problems. Unfortunately, the concept of rejection sensitivity was not mentioned in TEMPO. Additionally, it has also been observed that dating websites may be associated with low self-esteem and negative self-image following rejection (Strubel and Petrie, 2017). Rydahl et al. showed that individuals with a history of bipolar disorder were more likely to engage in online dating (Rydahl et al., 2021). Moreover, it is difficult to know the exact causal pathway of this association. Distress may predict online dating. The literature supports this idea (Coduto et al., 2020; Rydahl et al., 2021; Coffey et al., 2022; Mennig et al., 2022) even though the exact nature of the relationship has not been proven. However, some studies indicate that among individuals with a history of affective disorders, 49% reported that the use of online dating during depressive episodes aggravated their symptoms (Rydahl et al., 2021). This suggests that the relationship between the use of online dating sites and symptoms of anxiety and/or depression could be bidirectional and should be studied in longitudinal designs to confirm or refute it.

### *Limitations and strengths*

Prior to interpreting the findings, we need to mention several possible limitations. First, due to the design of TEMPO cohort and selective attrition, there is an over-representation of women and individuals with high socioeconomic positions among TEMPO participants. However, in our study, around half of the video game users played several times per week, which is similar to the results reported in 2015 by the French National Center for Cinema and Animated Image (Centre National du Cinéma et de l'image animée, 2015). Additionally, the average of 1 h and a half spent on social media daily was consistent with other national statistics (Gaudiaut, 2022), making our results generalizable to the French population. In 2013, approximately 16–21% of French adults aged 31–40 connected to dating applications (Bergström, 2016) which is close to the percentage of TEMPO participants who had sexual intercourse with someone met on a dating website (21%). Moreover, the levels of psychological difficulties among TEMPO cohort participants are comparable to national estimates, making it an appropriate sample to study the associations between screen use and mental health.

Second, in the literature, other tools are generally used to assess video game use, such as the Problem Video Game Playing scale, or

**Table 3.** Associations between screen use and symptoms of anxiety and/or depression. (TEMPO cohort study, France, 2018; logistic regression analyses, Odds-ratio (OR) and 95% confidence interval (95% CI), N = 853)

| | Unadjusted OR (95% CI) | *p*-value | Adjusted[a] OR (95% CI) | *p*-value |
|---|---|---|---|---|
| **Frequency of video game use** | | | | |
| Never | 1 | – | 1 | – |
| 1 to 4 times per month | 0.63 (0.32; 1.24) | 0.176 | 0.62 (0.28; 1.37) | 0.234 |
| Multiple times per week | 1.25 (0.73; 2.16) | 0.417 | 1.52 (0.80; 2.88) | 0.205 |
| **Time spent on social media per day** | | | | |
| Not at all | 1 | – | 1 | – |
| 2 h or less | 1.45 (0.61; 3.45) | 0.406 | 1.16 (0.39; 3.43) | 0.789 |
| More than 2 h | 1.41 (0.48; 4.11) | 0.534 | 0.47 (0.12; 1.79) | 0.265 |
| **Online dating leading to sexual relations** | | | | |
| No | 1 | – | 1 | – |
| Yes | 2.31 (1.38; 3.85) | 0.001 | 2.16 (1.15; 4.06) | 0.016 |

[a]Adjusted for gender, living circumstances, socioeconomic index, prior history of symptoms of anxiety and/or depression; number of close family members; number of close friends and need for more help from your partner, relatives and friends.

Internet Game Disorders Scale or time spent on video games, mainly conducted among adolescents (Tejeiro Salguero and Bersabé Moran, 2002; Tejeiro et al., 2016; AlSayyari and AlBuhairan, 2018). It would have been preferable to use these tools rather than asking questions about social media and internet use. Nevertheless, it provides an idea of the use of these media, even if comparisons with other existing studies are challenging.

Third, the models were not adjusted on physical activity because no data were available on this topic. It has been shown that physical activity enhances a wide range of affective wellbeing, including mental health, particularly in older people (Chen et al., 2022). Additionally, engaging in physical activity has been found to be a preventive factor for internet addiction (Sayili et al., 2023). Nevertheless, we took into account the time spent on social media per day, which could serve as a proxy of time spent on physical activity, as problematic social media use may be negatively associated with physical activity (Ren et al., 2022).

Our study has many strengths, which counterbalance the cited limitations. The longitudinal nature of the cohort reduces recall bias concerning the history of symptoms of anxiety and/or depression, which were assessed using data from previous waves of this cohort. Additionally, symptoms of anxiety and/or depression were assessed using a valid scale (Achenbach et al., 2003). Furthermore, we accounted for an important cofactor that is often overlooked in other studies – the history of same-sex sexual relations, sexual minorities being more likely to be current online dating users (Castro et al., 2020). Our regression models were adjusted for this factor as well as for social support and the feeling of needing more help.

The originality of this study lies in the studied population. Indeed, the patterns of screen use by adults aged 25 and over and their consequences are less well-known than among adolescents and young adults, even though adults also tend to use screens extensively – both professionally and recreationally – and are at risk of symptoms of anxiety and/or depression. Furthermore, to the best of our knowledge, ours is one of the few studies that have examined ODLSR as a possible correlate of psychological distress among adults.

## Conclusions

Our study extends the knowledge gained from research conducted among adolescents and contributes to a deeper understanding of this less-studied adult population. However, further research will be necessary to establish the direction of this association and the causal pathway. Additionally, it will aid in better managing individuals who suffer from mental health disorders and engage with these applications, as this can help them build self-confidence, improve self-image, and reduce fears of approaching others face-to-face. Moreover, the consequences of such use can be disastrous (Sparks et al., 2022). Future studies should investigate the determinants of using online meeting websites, including rejection sensitivity and specific mental health disorders such as low self-esteem and severe mental health conditions, and their relationship with the occurrence of psychological difficulties in longitudinal studies to establish causality.

**Open peer review.** To view the open peer review materials for this article, please visit http://doi.org/10.1017/gmh.2024.2.

**Supplementary material.** The supplementary material for this article can be found at http://doi.org/10.1017/gmh.2024.2.

**Data availability statement.** Due to the personal questions asked in this study, research participants were guaranteed that the raw data will be remain confidential. On reasonable request including standards for General Data Protection Regulation data can be accessed, please send an email to cohort.tempo@inserm.fr. Anonymized data can only be shared after explicit approval of the French national committee for data protection for approval (Commission Nationale de l'Informatique et des Libertés, CNIL).

**Acknowledgements.** The authors thank the TEMPO study participants who provided data for this project.

**Author contribution.** **Maria Melchior** conceptualized, designed the study and found funds. **Irwin Hecker** and **Solène Wallez** coordinated administratively the study, conducted the data collection and the investigations. **Maria Melchior** and **Murielle Mary-Krause** designed the methodology and statistical analysis protocol. **Maria El Haddad** conducted the statistical analyses under the supervision of **Maria Melchior** and **Murielle Mary-Krause**. **Maria El Haddad** wrote the first draft of the manuscript and all authors contributed to and have approved the final manuscript.

**Financial support.** The TEMPO cohort was supported by the French National Research Agency (ANR), the French Institute for Public Health Research-IReSP (TGIR Cohortes), the French Inter-departmental Mission for the Fight against Drugs and Drug Addiction (MILDeCA), the French Institute of Cancer (INCa) and the Pfizer Foundation.

**Competing interest.** The authors declare that they have no known competing financial interest or personal relationships that could have appeared to influence the work reported in this paper.

**Ethics statement.** The TEMPO cohort received approval of bodies supervising ethical data collection in France, the Advisory Committee on the Treatment of Information for Health Research (*Comité consultatif sur le traitement de l'information en matière de recherche dans le domaine de la santé*, CCTIRS) and the French computer watchdog authority (*Commission Nationale de l'Informatique et des Libertés*, CNIL, no. 908163).

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
