## [Reviewer Report]

Dear Editor, 

We would like to submit our paper, entitled The association between the use of video games, social media and online dating sites, and the anxiety-depression among adults over 30 for publication consideration in Global Mental health.

People tend to spend more time in front of their screens, even more during the COVID-19 pandemic, which can have repercussions on their social life, physical and mental health. This topic has mainly been studied in adolescents. So, our study tested associations between use of video games, social media and online dating websites, and anxiety-depressive disorders among adults over 30. Our study, conducted among a sample of adults drawn from the ongoing TEMPO cohort study, shows a significant association between online dating and symptoms of anxiety and depression, even after accounting for socio-demographic characteristics including same-sex sexual relations and preexisting psychological difficulties. This result is important to better target prevention campaign against internet and social media use, but also in order to better take care of people with mental health troubles using such apps.

We believe that our findings will be of interest to readers of Global mental health.

The authors declare no conflict of interest and all authors have approved the final version of the article. The content of the manuscript has not been published, or submitted for publication elsewhere. 

Thank you very much for considering our research for publication in your journal.

Sincerely Yours

---

## [Reviewer Report]

GMH Review of paper “The association between the use of video games, social media and online dating sites, and the anxiety-depression among adults over 30”

General

This paper describes the results of a large cohort study named TEMPO that analysed the associations between videogame use, social media use, use of dating apps on the one hand, and symptoms of anxiety and depression on the other hand in a France cohort sample.

The topic has potential relevance to mental and wellbeing following the COVID pandemic, and increases in screen use due to lockdown restrictions is a worldwide phenomenon. Therefore, it might be a topic of global relevance. That said, the paper was submitted to GMH, which as a journal is focused strongly on mental health across the globe. Therefore, the introduction would strengthen if the results would be placed in a global context, explaining the background evidence from studies across the world, including also studies from countries in non-Western settings and in the global south. Also, it would be good if the discussion also gets a more global focus.

Further, the paper in its current has important limitations that dampened my enthusiasm while reading. Because there are quite some language errors throughout, unfinished sentences and redundancies, the paper would benefit from very careful proof reading by the authors, ideally by a colleague who is a native speaker. Also, the paper would benefit from restructuring, especially the discussion section, which is a hard read because it contains redundancies and is very wordy.

Below comments per section are provided.

Title:

-It is stated in the title but also throughout the manuscript that your sample consists of adults of over 30, whereas in the Results section (line 213) it is surprisingly mentioned that adults of 25-44 were included? This is a major issue, and should be clarified cq. corrected throughout the entire paper.

-You use the rather uncommon construct anxiety-depression. It is better to refer to “symptoms of anxiety and depression”, or “psychological distress” (which usually consists of anxiety and depression). Here and there you also use internalizing symptoms, which is not often used in the context of adults. Please be consistent and use a single common description.

Impact statement:

-Lines 14-20: the sentence “Thus…. such apps” is grammatically incorrect (“being to assess”), very long and should be shortened or changed to two sentences.

Abstract:

-Methods: in addition to sex also provide mean age and age range.

-Results: what was the proportion of people scoring above the cut-off for symproms af anxiety and depression?

-Lines 4-43: In the conclusion section, the authors suggest that prevention campaigns should target internet and social media use. That is strange, since no association was found between that and with symptoms of anxiety and depression. Furthermore, since this is a cross-sectional study, no causal inferences can be made between the use of specific social media and websites, and the occurrence of mental health symptoms. Therefore, it is premature to make strong suggestions regarding prevention campaigns.

Introduction

-line 53: you refer to research on internet use among French adults aged 25-39, but your study is on adults aged 30 and older (although this is not entirely clear)? Could you also find evidence for adults? And although Statista is commonly used by researchers and journalists throughout the world, we cannot rely on its statistical data. Would you have any data from research articles?

-the introduction focuses very much on adolescents, probably because that is where the bul of the evidence lies concerning the detrimental effects of internet use. Rather than focusing on the adolescent population here, previous research around internet use (dating behavior, gaming, and social media use) should be described as well.

-line 65: a reference is needed after “life”.

-line 68: the sentence …, with different intentions, content and psychological intent” reads poorly and is not clear. What intent(ions) and content are you referring to?

-line 82: in what age group was the Marino et al (2018) study carried out?

-lines 88-92: “When a study exist (?) …. increased”: grammatically incorrect and very lengthy, please rephrase

-line 94: internalizing symptoms: use consistent term for symptoms of anxiety and depression, for example, psychological distress.

-

Methods

-Line 102: TEMPO is a longitudinal study but you decided to use only data form the 2018 time point. Why? In the discission you mention several occasions that longitudinal studies should be carried out, so why did you not use other timepoints in your analysis?

-Lines 123-128: please mention the psychometric properties of the ARS.

-Lines 130-136 describes “Indeed, in the literature, other tools are generally used to assess video game use such as the Problem Video Game Playing scale (PVP), or Internet Game Disorders Scale (IGDS9), or time spent on video games, mainly conducted among adolescents.”This is not very relevant for the methods section, but I suggest to use this text when describing the limitations of the way you measured your variables of interest in the discussion.

-Lines 140-142: it is not clear whether you divided time spent on video gaming into two or three categories.

-Lines 156-160: You measure online dating with the question: “Have you ever had sexual relations with someone you met online on a dating website?" but online dating is not always sexual in nature. People may be just in the “talking stage” or “get to know each other phase”. Therefore, you should consider to give your construct a more appropriate and specific name, reflecting “online dating resulting from sexual relations” or using a similar description.

-line 158: “do not wish to answer” is considered as a “no”. This is a major problem, since it is more likely that these people would actually engage in online dating but are ashamed to answer. This is even more likely because you inquire about sexual relations resulting from it which may be associated with stigma. This variable should be coded as a missing variable and your data need to be re-analyzed with this variable recoded as such.

-line 167 (covariates): Why do you distinguish between under 30 years and above 30 years if you intend to include only people aged 30 years and above?

-line 182: You mention that you measured history of anxiety and depressive disorders. Since the ASR is described to be a self-report instrument, no disorders can be diagnosed with it, but rather the history of increased levels of psychological distress. It has been found that self-report instrument generally overestimate the presence of a disorder. Please also specify at what wave(s) this was measured and how

-line 200 (statistical analysis): what type of bivariate analyses did you carry out (Pearson correlations)? And what p-value did you consider to indicate statistical significance?

Results

-Line 230: “level of symptoms of anxiety-depression., as However…” Please correct

-Line 233: “does” should be “did”

Discussion

-the discussion is particularly hard to read. It is lengthy and it mentions several issues multiple times in different wording. In all, the text can be made more concise. The text on limitations and strengths could be greatly reduced. You should only mention 2-3 sentences describing strengths, and not more than one paragraph describing limitations.

-the discussion does not pay much attention to the fact that you in fact measured prolific sexual behaviors in part of your sample (engaging in sexual relations using online dating apps), you may add a few sentences on the association between these behaviors and mental health (anxiety and depression) found in previous research.

-Despite shortening the limitations section, an additional limitation that should be mentioned is how problematic internet use was measured. This research field is booming, and instruments are developed and validated for standardized measurement.

-lines 246-248: in the first paragraph of the discussion section, you already describe clinical implications of your findings and future research directions. It is better to add a more elaborated paragraph with these implications for practice, and a separate paragraph with future research implications at the end of your discussion, for example just before the final conclusion.

-Lines 249-262: how do you explain not finding an association between video gaming and social media use, and psychological distress in your study? Did you measure these variables differently, do you think that your sample differs? This can be better explained.

-Line 255: “(BFAS) and “: delete “and”

-Line 258: you are suggesting that you studied middle aged adults, which is only partly the case. Usually, middle aged adults are people between the age range of 40 to 60 whereas your sample was 25-44.

-Line 260: the sentence “Whether or not video …. verified in future studies” is confusing, since this was the aim of your study? Why do you suggest that another study needs to be carried out on the same topic? You sample size was large, so it may not be worthwhile to do another study on the exact same topic that may also not find any relation between video gaming and social media use and psychological distress?

-Lines 281-282: you mention that the relation between online dating and psychological distress is likely to be bidirectional. Would it be possible that it is not bidirectional, but that distress only predicts online dating, and not the other way around? I don’t think that your analyses can rule this hypothesis out. Further, the fact that you can’t draw any conclusions regarding causality between these two variables, is mentioned again under the limitations. It would be good to delete it under the limitations.

-Line 358: if it is in fact distress that causes people to seek for a partner online, it may not be the right target for a prevention campaign. It is tricky to arrive at clinical suggestions in the absence of causal proof. Also, the sentence “…. target prevention campaign….” is not grammatically correct.

-Line 361: you suggest that new longitudinal studies need to be carried out. Can’t you use future waves of the TEMPO study to answer these questions?

---

## [Reviewer Report]

The study’s analysis of 2018 TEMPO cohort data revealed a significant association between online dating and anxiety-depression symptoms, particularly among women. By focusing on a crucial demographic—middle-aged individuals aged 30 and above—this research provides valuable insights into the associations between social media, online dating, and anxiety-depressive disorders in adults over 30. Since previous studies have predominantly concentrated on adolescents, this study is particularly significant. Its findings have important implications for developing prevention campaigns targeting internet and social media use (specifically, online dating) in the 30+ age group and providing better care for those experiencing anxiety-depression symptoms.

The literature review is comprehensive, with references that are relevant to the topic, covering both historical literature and more recent developments. The few existing studies on online dating, social media usage, and depression are mostly listed and reviewed in the literature, demonstrating the review’s thoroughness.

The manuscript is well-written and organized. The use of the Adult Self-Report (ASR) is appropriate, and the variable regarding time spent on social media is well justified.

The statistical analysis was robust, and the findings were reasonable, given the study’s limitations (e.g., sample size and origins). Notably, the researchers addressed these limitations to ensure an accurate interpretation of the results. However, it would be beneficial for the author to provide more information on the TEMPO cohort, including the exact age range of the participants.

---

## [Reviewer Report]

Dear Editor, 

Further to your response about the manuscript entitled “The association between the use of video games, social media and online dating sites, and the anxiety-depression among adults over 30”, reference GMH-23-0023, we would like to thank you for your interest in our manuscript. According to the reviewers’ comments, we have revised the manuscript and responded point-by-point to their comments. We also notified in red modifications in the manuscript, Tables, Highlights and Supplementary Material. We hope that our revised version is satisfactory.

Thank you very much for considering our research for publication in your journal.

Sincerely Yours

Murielle Mary-Krause

---

## [Reviewer Report]

I appreciate the significant improvements made to the manuscript since the last review. However, I would like to highlight two primary concerns that must be addressed.

Concern 1:

In your response to a previous reviewer’s comment regarding line 158, you have classified the responses, “do not wish to answer,” as a “no”. Your justification was that only four participants selected this option and that this small number did not warrant a separate category, nor did it significantly alter your results. However, this decision is problematic.

The standard practice in data analysis is to dichotomize responses into “yes” and “no”, treating any non-answers as missing data or categorizing them separately. The choice to not answer is not equivalent to a negative response. While it’s true that the current size of this group (n=4) is small, the methodology should be robust enough to handle larger numbers. If it were n=400, for example, the impact could be significant. I would therefore advise treating these responses as missing data, focusing only on genuine yes/no responses.

Concern 2:

In the discussion section (Page 16), you have included a substantial new paragraph on the concepts of rejection sensitivity, low self-esteem, and how these factors relate to the use of online dating websites. References to studies by Blackhart et al., 2014; Hance et al., 2017; Strubel and Petrie, 2017; and, Rydahl et al., 2021 have been included in this discussion.

However, these concepts and references are not present in the literature review, and it appears that the TEMPO study data you provided in the study does not include these specific considerations. To maintain consistency and offer a comprehensive review of the topic, it would be beneficial to introduce these concepts in the literature review section if you intend to discuss them later. Alternatively, if the current dataset does not include these factors, they might be better placed as recommendations for future research.

---

## [Reviewer Report]

Pr Gary Belkin

Editor in Chief

Global Mental Health

Paris, November 2nd, 2023

Dear Editor, 

Further to your response about the manuscript entitled “The association between the use of video games, social media and online dating sites, and the anxiety-depression among adults over 30”, reference GMH-23-0023, we would like to thank you for your interest in our manuscript. According to the reviewer’s comments, we have revised the manuscript and responded point-by-point to his.her comments. We also notified in red modifications in the manuscript, Tables, Highlights and Supplementary Material. We hope that our revised version is satisfactory.

Thank you very much for considering our research for publication in your journal.

Sincerely Yours

Murielle Mary-Krause

---

## [Reviewer Report]

In reviewing the revised paper, I note the authors‘ revision in the treatment of ’do not wish to answer‘ responses. Initially, equating these responses with a ’no' could potentially misrepresent the data. The updated approach (P9-10), which refrains from interpreting these as negative responses, seems more appropriate and in line with sound research practices. The re-analysis of the data to account for this change is also acknowledged (P12-13/24-25). I am satisfied with this methodological revision.